

# BSEFNet: bidirectional self-attention edge fusion network salient object detection based on deep fusion of edge features

Gan Gao, Yuanyuan Wang, Feng Zhou, Shuaiting Chen, Xiaole Ge and Rugang Wang

School of Information Technology, Yancheng Institute of Technology, Yancheng, JiangSu, China

## ABSTRACT

Salient object detection aims to identify the most prominent objects within an image. With the advent of fully convolutional networks (FCNs), deep learning-based saliency detection models have increasingly leveraged FCNs for pixel-level saliency prediction. However, many existing algorithms face challenges in accurately delineating target boundaries, primarily due to insufficient utilization of edge information. To address this issue, we propose a novel approach to improve the boundary accuracy of salient target detection by integrating salient target and edge information. Our approach comprises two key components: a Self-attentive Group Pixel Fusion module (SGPFM) and a Bidirectional Feature Fusion module (BFF). The SGPFM extracts salient edge features from the lower layers of ResNet50 and salient target features from the higher layers. These features are then optimized using a self-attentive mechanism. The BFF module progressively fuses the salient target and edge features, optimizing them based on their logical relationships and enhancing the complementarities among the features. By combining detailed edge information and positional target information, our method significantly enhances the detection accuracy of target boundaries. Experimental results demonstrate that the proposed model outperforms the latest existing methods across four benchmark datasets, providing accurate and detail-rich salient target predictions. This advancement marks a significant contribution to the development of the field.

# INTRODUCTION

Salient object detection (SOD) aims to mimic the human visual attention mechanism to identify the most salient regions in an image and accurately localize critical foreground information. As an important preprocessing step in the field of computer vision, SOD is widely used in several visual tasks such as image retrieval (*Jain et al., 2023*), visual tracking (*Chen et al., 2023*), medical image segmentation (*Santhirasekaram et al., 2023*), photo synthesis, and collaborative saliency detection(*Fan et al., 2021*). In addition, research in video saliency target detection, RGB-D saliency target detection, and weakly supervised saliency target detection is also gaining momentum.

Traditional saliency target detection models are mainly based on a bottom-up approach to classify the saliency of samples by utilizing different underlying visual features. Significant

Corresponding author
Yuanyuan Wang, wyy@ycit.edu.cn

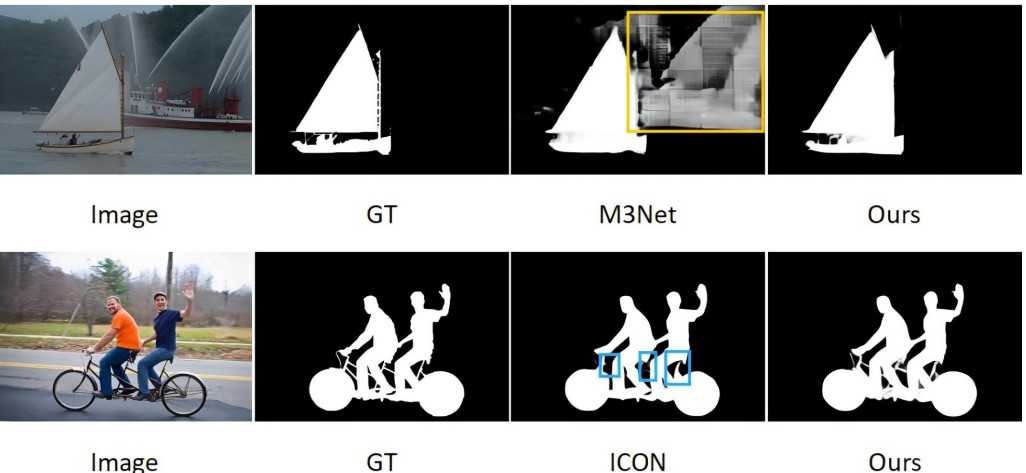

**Figure 1** In the existing SOD model, background objects are incorrectly predicted as salient objects in the case of M3Net (yellow box), and some salient details are missing in the case of ICON (blue box).

progress in SOD has been made with the widespread use of convolutional neural networks (CNNs) (*LeCun et al., 1998*). CNNs have performed well in several image tasks, such as target detection, semantic segmentation (*Lu, de Geus & Dubbelman, 2023*), and edge detection. For SOD, CNNs introduce new ideas and show significant improvements in many studies. Due to its multi-stage and multi-scale nature, CNN is able to accurately capture the most salient regions without prior knowledge. In addition, the multi-stage feature helps CNNs to locate the boundaries of salient objects more accurately even in the face of situations such as shadows or reflections. As a result, CNN-based SOD methods have set records on almost all existing datasets and become mainstream methods.

With the rise of fully convolutional neural networks (FCNs), SOD methods based on deep learning are further developed to achieve more effective feature representation. Inspired by FCNs, more and more pixel-level SOD methods have emerged. However, despite the excellent performance of CNNs in SOD, existing methods still face two major challenges: (1) partial missing of salient targets or incorrectly predicting backgrounds as foregrounds; and (2) lack of fine edge information in the predicted saliency results. As shown in Fig. 1.

To solve these problems, we propose the Bidirectional Self-Attention Edge Fusion Network (BSEFNet). BSEFNet improves the accuracy of feature representation and edge details by introducing the Self-Attention Group Pixel Fusion module (SGPFM) and Bidirectional Feature Fusion (BFF); BSEFNet improves the accuracy of feature representation and edge details. Our main contributions are as follows:

(1) SGPFM is designed for optimization within a network to enhance the representation of salient target features and edge features. The module generates groups of features with different receptive fields *via* the Self Attention Group Pixel Module (SGPM), while the Self Attention Group Fusion Module (SGFM) achieves the optimal combination.

(2) A BFF module is introduced to leverage the logical relationship between salient edge features and salient object features by cross-optimizing them. Through top-down optimization, this module enables the multi-scale features of salient objects to gradually acquire more details while suppressing the background interference of salient edges.

(3) Numerous experiments demonstrate that our proposed BSEFNet performs better compared to the current state-of-the-art algorithms on four classical benchmark datasets. The framework is not only capable of accurately segmenting salient targets, but also of accurately detecting salient edges.

In conclusion, BSEFNet effectively solves the two main challenges in SOD by combining the self-attention mechanism and bi-directional feature fusion, which significantly improves the accuracy and fineness of detection.

# RELATED WORK

Over the past few years, the development of conventional techniques involved early algorithms (*Wang et al., 2019*) for the recognition of salient targets in images. These early algorithms usually involved manual extraction of features and visual cues such as color, texture, position and luminance. Traditional techniques usually require a priori knowledge in detecting salient targets and do not adequately consider contextual semantic information, resulting in low detection performance. With the rise of CNNs, more and more models have begun to employ deep learning for salient target detection. Although deep learning-based methods have made some progress (*Chen et al., 2020*; *Pang et al., 2020*; *Liu et al., 2021*), how to effectively combine spatial information and contextual semantic details is still a key issue.

## Hierarchical feature fusion model

*Ren et al. (2020)* introduced a local and global context fusion approach to provide a more effective solution for salient target detection, and experimentally verified its superior performance, providing valuable insights into the field. *Wei et al. (2020)* proposed a label decoupling framework, which solves the feature confusion problem of the traditional approach by separating different salient features, and significantly improves the detection performance. The framework provides new perspectives and methods for research in this area. *Zhu et al. (2019)* developed a feature aggregation method that combines inflated convolution and attention mechanisms, which significantly improves the performance of salient target detection, especially in complex scenes. This research provides a new technical approach for the field of saliency detection and promotes the development of this field. *Zhang, Shi & Zhang (2020)* provide a new solution for salient target detection by introducing the attention mechanism and boundary guidance technique, which effectively improves the detection performance of the model in complex scenes. This research makes an important contribution to the development of this field and has a wide range of application potential. *Hu et al. (2020)* significantly improved the accuracy and robustness of salient target detection by introducing the innovative concept of spatial decay context, which provides new methods and ideas for research and application in this field. Several studies have introduced Transformer (*Dosovitskiy et al., 2021*) to dense prediction tasks. Due

to Transformer's ability to quickly establish long-term dependencies, Transformer-based SOD methods (*Yun & Lin, 2022*; *Tang et al., 2022*) perform well in localizing salient regions compared to CNN alternatives. However, using only global self-attention may result in the loss of a large amount of local details.

To overcome these problems, this paper proposes SGPFM, which is able to autonomously optimize the in-layer information and enhance the characterization of salient and edge features by employing a group-optimized fusion strategy.

## Boundary guided models

In recent research on supervised learning-based segmentation models, some models have been trained to embed edge-related knowledge into the SOD task by generating a saliency graph that preserves boundaries using labeled data. In order to enhance the boundary of the saliency graph (*Feng, Lu & Ding, 2019*), the loss function is adjusted along with the boundary loss, and an attention-based feedback method is used to evaluate the object structure. In EGNet (*Zhao et al., 2019*), edge information is extracted from the low-level features of the backbone network by modifying the edge and global position details of the fused object with target features. In order to be more accurate and maintain the saliency map of edges, *Wu, Su & Huang (2019)* employs an edge detection mechanism and combines it with a target detection algorithm. BASNet (*Qin et al., 2019*) estimates the saliency map through an encoder–decoder network and improves it by employing residual refinement techniques.It is worth noting that these edge-guided models employ different optimization strategies. Some use only edge information to refine the contours of salient objects, but such edge information may contain too much background interference and appear coarse. Some other edge-guided models use a bidirectional optimization strategy but fail to take full advantage of the relationship between edge information and saliency information.

To solve this problem, this paper proposes a BFF module, which is designed based on the relationship between the two, and can transfer information in both directions between the SOD task and the SED task to realize the top-down gradual acquisition of more complementary details of multilevel features.

## METHOD

### Overall framework

Our proposed BSEFNet adopts an encoder–decoder architecture and comprises four main components: the backbone ResNet50 (*He et al., 2016*), DASPP (*Yang et al., 2018*), SGPFM, and BFF. An overview of our network is illustrated in Fig. 2.

In the encoder section of our model, we employed a combination of a backbone network, a Dense Atrous Spatial Pyramid Pooling (DASPP) module, and five SGPFMs. We chose ResNet50 as the backbone network because its deep architecture effectively addresses the gradient vanishing problem in deep networks through the introduction of residual blocks, which enhances training stability and convergence speed. Additionally, ResNet50 is renowned for its ability to efficiently extract rich feature representations and has been extensively validated across various computer vision tasks. This makes it ideal for

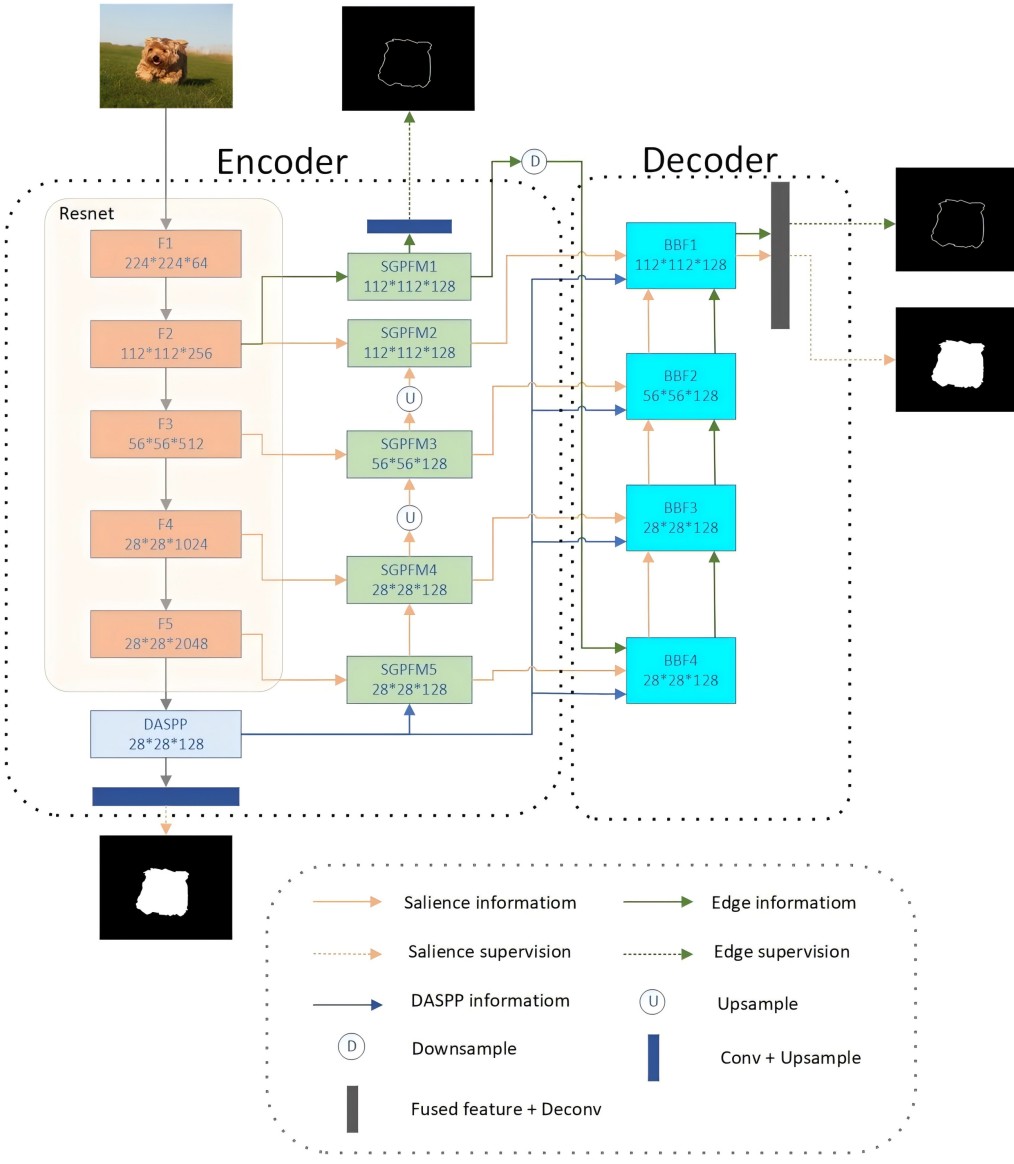

**Figure 2 Schematic diagram of the network structure of the proposed BSEFNet.** The overall framework consists of ResNet50, a dense spatial pyramid pooling model (DASPP), five SGPFMs and BFF modules. The second layer of our backbone extracts the edge information and then optimizes both information in a top-down interaction.

providing robust feature extraction and an efficient training process when constructing complex encoder–decoder architectures. We selected DASPP for its capability to extract multi-scale features effectively by employing dilated convolutions, which improve the model's robustness to various object sizes and backgrounds by expanding the receptive field without increasing the computational load.

To optimize the network, we removed the fully connected layers from the final set of ResNet50 blocks and labeled the remaining convolutional blocks as F1 through F5. Given

the high computational cost associated with fully connected layers, we retained all the convolutional layers of ResNet50 as our backbone to efficiently extract salient features across multiple scales. Following the F2 block, we introduced an SGPFM to extract rough foreground edge features, which are then supervised using salient edge labeling. Alongside the F1 through F5 blocks, we implemented four additional SGPFMs to extract multi-scale salient target features. Moreover, at the apex of the F5 block, we applied the DASPP module, with expansion rates set to 1, 6, 12, and 18, to extract global saliency features and supervise them with segmentation labeling. These extracted features are then passed to the decoder section for further optimization.

The decoder section utilizes a progressive optimization architecture for multi-scale feature aggregation and cross-optimization of edge and object features. It comprises four BFF modules, each with a similar structure. For instance, BFF3 has four inputs: salient object features and salient edge features from BFF4, salient object side features from SGPFM4, and global semantic saliency features sampled from DASPP. BFF4 fuses the salient object features from these four inputs and cross-optimizes them with the salient edge features to produce refined features. Notably, we discard the feature from F1 due to its high content of non-significant information. The final prediction graph is generated by BFF1 and is supervised by saliency and edge labeling. This structure's advantage lies in its multi-level and multi-scale optimization of saliency features, which significantly enhances the model's performance in the saliency target detection task.

## Self attention group pixel fusion module

The SGPFM module aims to autonomously optimize features within the convolutional layer and enhance feature representation. It comprises two key components: SGPM and SGFM. SGPM accepts inputs from backbone blocks F2-F5 and generates a set of features with progressively increasing receptive fields in a top-down feature fusion manner. On the other hand, SGFM efficiently combines features grouped in different receptive fields. Each output of SGPFM1-SGPFM5 has the same number of channels. Compared to other feature refinement methods in the SOD task, SGPFM can produce more representative features with the assistance of SGPM and SGFM, as depicted in Fig. 3.

For SGPM, the objective is to achieve self-optimization of features within layers. Initially, the input of SGPFM is divided into a set of features, represented as $\{X_j\}, j = 1, \{\ldots, N\}$. Feature $X_1$ is merged with the output of SGPFMi+1 to incorporate additional semantic information. Subsequently, each $X_j$ is passed through the self-attentive convolutional layer for feature refinement. Since $X_{j+1}(j = 1, \ldots, N-1)$ is combined with $X_j$, the portion of $X_{j+1}$ undergoes more convolutional operations, resulting in a progressive increase in the receptive field of the grouped features of the output (denoted as $F_j, j = 1, \ldots, N$). Overall, SGPM achieves semantic enhancement of intra-layer features.

The SGFM is devised to amalgamate the grouped features output by SGPM. As mentioned earlier, the grouped features exhibit progressively increasing receptive fields. Features grouped with larger receptive fields encapsulate more global information, whereas those with smaller receptive fields contain finer local details. Acknowledging the complementary nature of local and global information, we design a residual structure

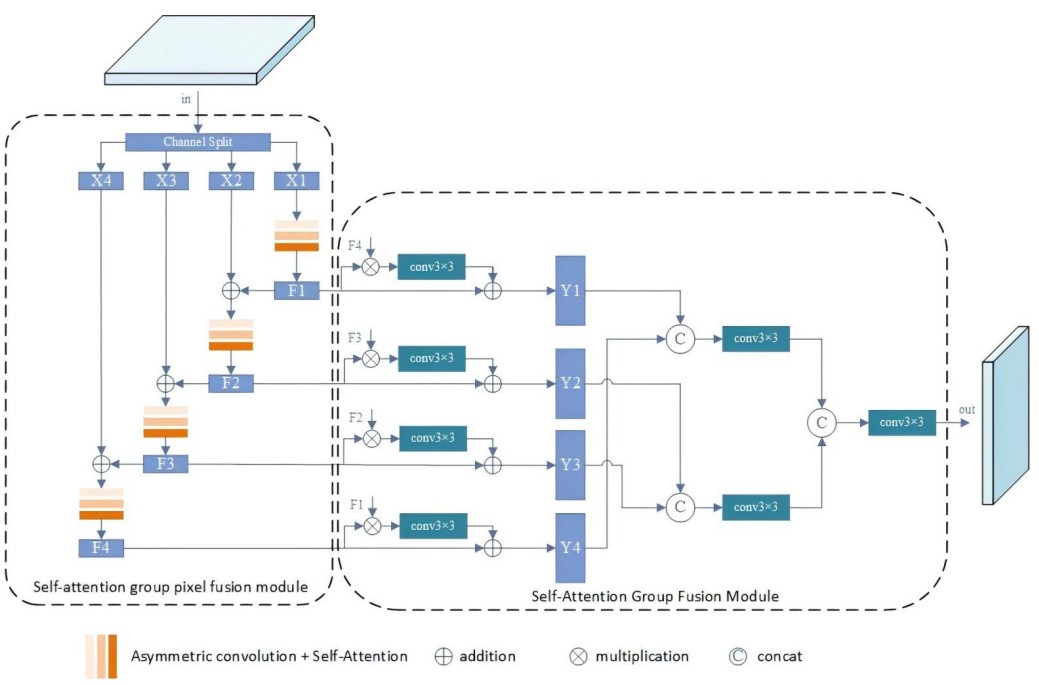

**Figure 3** **Illustration of the SGPFM.** The SGPFM consists of two key components: SGPM and SGFM.

to integrate the grouped features with varying receptive fields. The general formula for SGFM is expressed as follows:

$$Y_i = F_i + \text{Conv}_{3\times3}(F_i * F_{N+1-i}), i = 1, \ldots, N. \tag{1}$$

In the general formula for SGFM, N represents the number of splits in a group, and the symbols + and * denote the average values obtained by summation and multiplication of elements, respectively. The grouped features $Y_i$ are subsequently concatenated to generate the final output.

## Bidirectional feature fusion module

We've developed the BFF module to facilitate bidirectional message passing between salient features and salient edge features.

As depicted in Fig. 4, three saliency features are initially combined by the fusion module and subsequently optimized in conjunction with edge features. By incorporating these BFFs in a top-down manner, the multi-scale edge and segmentation features gradually acquire additional details. The multi-scale saliency features are initially aggregated using the feature fusion module (FFM). For each BFFi, the global semantic salient features from DASPP are represented as $S_g$, the upper salient features from BFFi+1 are denoted as $S_u$, the current layer salient features from SGPFMi are denoted as S, and the output of the fusion module is labeled as $S_f$. This fusion process can be described as follows:

$$F_1 = relu\big(\text{Conv}_{3\times3}(S) * upsample(\text{Conv}_{3\times3}(S_u))\big) \tag{2}$$
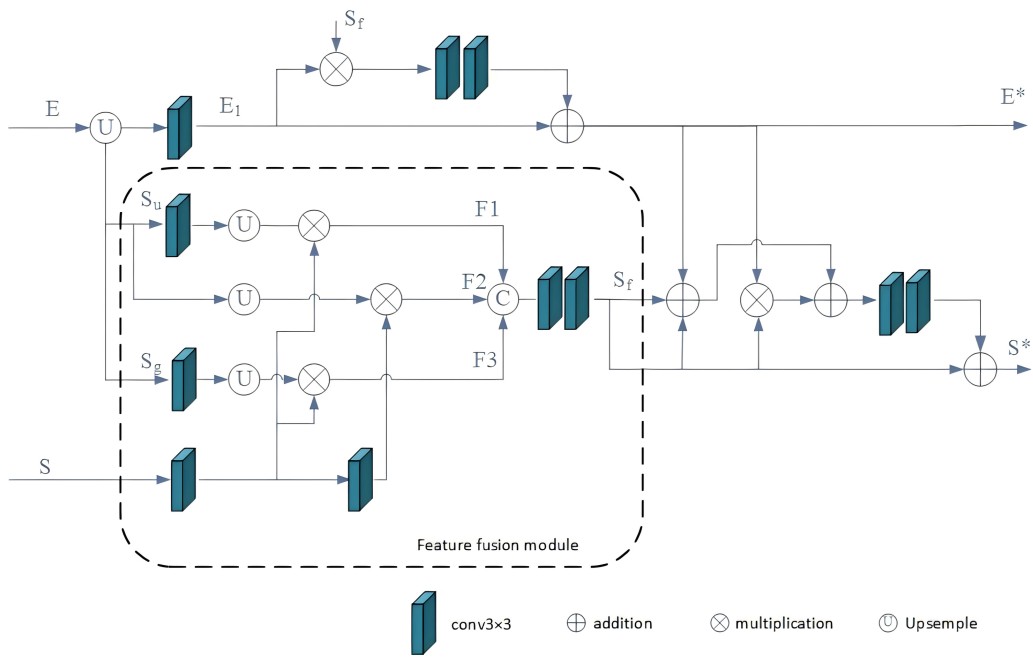

**Figure 4 BFF module.** The three salient features are first integrated by FFM and then cross-optimized with the edge features.

$$F_2 = relu\left(Conv_{3\times3}(Conv_{3\times3}(S)) * upsample(S_u)\right) \tag{3}$$

$$F_3 = relu\left(Conv_{3\times3}(S) * upsample\left(Conv_{3\times3}(S_g)\right)\right) \tag{4}$$

$$S_f = Conv_{3\times3}\left(Conv_{3\times3}\left(Concat(F_1, F_2, F_3)\right)\right) \tag{5}$$

where $F_1$, $F_2$, and $F_3$ represent intermediate features, the operation "Concat" signifies that the features are concatenated along the channel dimension. High-level features encapsulate abundant semantic information, enhancing the ability to localize targets and emphasize foreground objects. Consequently, this paper introduces a feature fusion algorithm to refine the current layer salient features by incorporating multi-layer salient features. This process aims to diminish background interference in the current layer features and augment semantic information. Subsequently, the fused saliency features undergo cross-optimization with edge features.

The cross-optimization component of BFF, grounded in the logical relationship between SOD and SED, is depicted in Fig. 3. In this context, for each BFFi, the edge feature from BFFi+1 is labeled as E, while the resulting edge feature and saliency feature are denoted as

$E^*$ and $S^*$, respectively. Accordingly, the process of cross-optimization can be elucidated as follows:

$$E_1 = Conv_{3\times3}(upsample(E)) \tag{6}$$

$$E^* = E_1 + Conv_{3\times3}\left(Conv_{3\times3}(E_1 * S_f)\right) \tag{7}$$

$$S^* = S_f + Conv_{3\times3}\left(Conv_{3\times3}\left(E^* + S_f - E^* * S_f\right)\right). \tag{8}$$

In the provided equation, $B_1$ represents the intermediate edge feature, and $S_f$ denotes the output of the fusion model. The operation $E_1 * S_f$ preserves the shared portion of salient and edge features. Additionally, the operation $E^* + S_f - E^* * S_f$ refines the edges of salient features. Specifically, the operation $E^* + S_f$ compensates for the contours of salient features, while the operation $-E^* * S_f$ removes the extra edge portion introduced by the addition. Following the top-down configuration of the BFF module, the salient target features gradually acquire finer edge information, effectively suppressing background interference in the salient edge features.

## Loss function

For efficient training, multi-supervision is used for both salient target detection and salient edge detection. We just use a binary cross-entropy (BCE) loss function to define the loss for the SED task. In order to efficiently train SOD and significant edge detection (SED), we adopt a multi-supervised strategy to fully utilize the information and improve the accuracy and robustness of the detection. For the SED task, we chose BCE loss function because it is suitable for dealing with binary classification problems, provides stable gradients, and has flexibility in dealing with positive and negative sample imbalances. Compared to other loss functions used for salient target detection, such as the Dice loss, which is mainly used to deal with unbalanced data, the IoU loss, which has high computational complexity, and the Focal loss, which requires tuning of more hyper-parameters, the BCE loss function is more straightforward and effective in the salient edge detection task. The BCE loss is expressed as follows:

$$L_{BCE} = \sum_{p\in\mathcal{P}, g\in\mathcal{G}} -[g\log p + (1-g)\log(1-p)] \tag{9}$$

where $\mathcal{P} \in [0,1]^{H\times W\times 1}$ denotes the prediction and $\mathcal{G} \in \{0,1\}^{H\times W\times 1}$ denotes the true value.

Unlike BCE loss, which operates at the pixel level, cross-entropy loss (CEL) (*Pang et al., 2020*) can incorporate global content and has been demonstrated to preserve consistency in foreground highlights. Consequently, a combination of BCE and CEL is employed as the loss function for the salient target detection task. The CEL loss is denoted as follows:

$$L_{CEL} = \frac{\sum(p-pg) + \sum(g-pg)}{\sum p + \sum g}. \tag{10}$$

In the provided equation, $p \in \mathcal{P}$, and $g \in \mathcal{G}$. To summarize, the marginal loss and the significance loss are denoted by:

$$L_{edge} = L_{BCE} \tag{11}$$

$$L_{sal} = \eth * L_{BCE} + (1 - \eth) * L_{CEL}. \tag{12}$$

The overall loss function is:

$$L = \lambda_1 L_{edge}^m + \lambda_2 L_{sal}^m + L_{edge} + L_{sal}. \tag{13}$$

In the provided equation, $L_{edge}^m$ and $L_{sal}^m$ represent the significance loss and edge loss in the middle layer, respectively. The symbol o, as defined in Eq. (12), balances the BCE loss and CEL loss in the saliency supervision. Additionally, $\lambda_1$ and $\lambda_2$, as defined in Eq. (12), balance the significance loss and edge loss in the intermediate layer, respectively. We set to o $= 0.8$ and $\lambda_1 = \lambda_2 = 0.6$.

# EXPERIMENTAL RESULT

## Experimental setup

The proposed BSEFNet is implemented using the public PyTorch toolkit, and all experiments are conducted on a single Nvidia-GTX 3090 GPU. We utilized ResNet50 for pre-training the backbone network, with subsequent fine-tuning of the BSEFNet network using the DUTS-TR dataset. The weights of the other convolutional layers are initialized using a normal distribution with a mean of zero and a standard deviation of 0.01.

In the SGPFM, we set specific hyperparameters for different modules: the number of segments $N = 2$ for SGPFM1 and SGPFM2, and $N = 4$ for SGPFM3 through SGPFM5. The batch size is set to 3, and the input image sizes are resized for data augmentation to $[448 \times 448]$, $[224 \times 224]$, $[112 \times 112]$, $[56 \times 56]$, and $[28 \times 28]$.

The model is trained using the Adam optimizer over 50 epochs. The initial learning rate is set to $2e-5$ and is decayed by a factor of 0.5 at epochs 14 and 22. To prevent gradient explosion, the gradient in the optimizer is clipped within the range of $[-0.5, 0.5]$. This training strategy ensures a stable and efficient learning process, contributing to the robust performance of the BSEFNet model.

## Data set

For the saliency detection task, the popular DUTS-TR (*Wang et al., 2017*) dataset was used to train our BSEFNet. DUTS-TE (*Wang et al., 2017*) and three other popular datasets HKU-IS (*Li & Yu, 2015*), ECSSD (*Yan et al., 2013*), and PACSALS (*Everingham et al., 2010*) were used to evaluate the performance of the model. DUTS (*Wang et al., 2019*) is the largest significant target detection dataset containing 10,553 images for training and 5,019 images for testing. Most of the images have significant variations in position and scale. The PASCAL-S dataset contains 850 images, all of which were selected from the PASCAL VOC dataset. ECSSD contains 1,000 natural and meaningful semantic images

with a variety of complex scenarios. These images were manually selected from the Internet. HKU-IS contains 4,447 images with high-quality annotations, many of which have multiple disconnected salient objects and low contrast.

## Evaluation indicators

To evaluate the quantitative performance, we used the mean absolute error MAE, F-measure, and precision recall (PR) aspects for comparison. We first threshold the prediction map to binary and compute the precision and recall value pairs for different thresholds. We compute the F-measure for different precision–recall pairs with Eq. (14):

$$F_\beta = \frac{(1+\beta^2)\text{Precision} \times \text{Recall}}{\beta^2 \times \text{Precision} + \text{Recall}}. \tag{14}$$

In line with prevalent methodologies (*Su et al., 2019*; *Perazzi et al., 2012*), we adopt $\beta^2 = 0.3$ as the metric for assessing precision over recall. The maximal F-measure and the mean F-measure are derived from the maximal and average F-measure values, respectively, computed across all exact recall pairs.

Furthermore, PR curves delineating various precision-recall pairs are plotted to juxtapose the efficacy of our algorithm against alternative methods across the four datasets. Eq. (15):

$$Precision = \frac{TP}{TP+TF}, Recall = \frac{TP}{TP+FN}. \tag{15}$$

The mean absolute error (MAE) serves as a prevalent metric for quantifying the disparity between predicted and true value maps. Herein, we denote the predicted significance map and the corresponding ground truth as P and G, respectively. The MAE score is calculated as follows. Eq. (16):

$$MAE = \frac{1}{W \times H} \sum_{x=1}^{W} \sum_{y=1}^{H} |P(x,y) - G(x,y)| \tag{16}$$

where W and H represent the width and height of the image, respectively.

## Performance comparison

To validate the effectiveness of our method, we compare the proposed BSEFNet with state-of-the-art methods in this section. The comparison includes various leading techniques such as AFNet (*Feng, Lu & Ding, 2019*), BASNet (*Qin et al., 2019*), MINet (*Pang et al., 2020*), U2Net (*Qin et al., 2020*), ITSD (*Zhou et al., 2020*), DSRNet (*Wang et al., 2020*), WSSA (*Zhang et al., 2020*), ICON-R (*Zhuge et al., 2022*), M3Net-R (*Yuan, Gao & Tan, 2023*), and GFINet (*Zhu, Li & Guo, 2023*). For a fair comparison, the saliency maps were either provided by the authors of these methods or generated using officially released pre-trained models.

As shown in Table 1, our model demonstrates strong performance compared to the state-of-the-art methods across most academic evaluation metrics. Figure 5 presents some visual comparison results with four other leading models, highlighting the superior saliency detection capabilities of our BSEFNet model.

In addition to the numerical comparisons presented in Table 1, we illustrate the PR curves and F-measure curves for select compared methods across the four datasets in Fig. 6.

**Table 1  Quantitative comparison of the state-of-the-art significance models on the four benchmark datasets in terms of maximum F-measure, average F-measure.** Bold and underline text indicate the best and second best performance, respectively, and the symbols "↓ /↑" indicate that the lower/higher the evaluation, the model is approximately better, and the indicator "-" indicates that the model is not available. Overall, the proposed BSEFNet has superior performance.

| Method | DUTS-TE | | | HKU-IS | | | ECSSD | | | PASCAL-S | | |
|---|---|---|---|---|---|---|---|---|---|---|---|---|
| | MAE↓ | MaxF↑ | MeanF↑ | MAE↓ | MaxF↑ | MeanF↑ | MAE↓ | MaxF↑ | MeanF↑ | MAE↓ | MaxF↑ | MeanF↑ |
| AFNet[19] | 0.045 | – | 0.785 | 0.036 | – | 0.905 | 0.042 | – | 0.886 | 0.070 | – | 0.797 |
| BASNet[19] | 0.048 | 0.859 | 0.796 | 0.032 | 0.928 | 0.896 | 0.037 | 0.916 | 0.855 | 0.077 | 0.862 | 0.779 |
| MINet[20] | 0.038 | 0.883 | 0.832 | 0.028 | 0.935 | 0.908 | 0.036 | 0.945 | 0.923 | 0.064 | 0.867 | 0.829 |
| U²Net[20] | 0.045 | 0.872 | 0.797 | 0.032 | 0.935 | 0.893 | 0.036 | 0.947 | 0.890 | 0.074 | 0.859 | 0.774 |
| ITSD[20] | 0.041 | 0.882 | 0.808 | 0.031 | 0.933 | 0.898 | 0.037 | 0.944 | 0.892 | 0.066 | 0.870 | 0.785 |
| WSSA[20] | 0.063 | 0.789 | 0.744 | 0.046 | 0.884 | 0.864 | 0.061 | 0.889 | 0.870 | 0.092 | 0.809 | 0.774 |
| DSRNet[21] | 0.043 | 0.883 | 0.796 | 0.035 | 0.933 | 0.893 | 0.042 | 0.908 | 0.941 | 0.067 | 0.874 | 0.819 |
| ICON-R[22] | 0.037 | – | 0.836 | 0.029 | – | 0.902 | 0.032 | – | 0.918 | 0.064 | – | 0.818 |
| M³Net-R[23] | 0.037 | – | 0.849 | **0.027** | – | 0.913 | **0.030** | – | 0.919 | **0.061** | – | 0.827 |
| GFINet[23] | 0.038 | – | **0.890** | 0.028 | – | **0.939** | 0.032 | – | **0.948** | 0.066 | – | 0.876 |
| Our | **0.035** | **0.888** | 0.876 | 0.029 | **0.936** | 0.918 | 0.034 | **0.951** | 0.935 | 0.063 | **0.888** | **0.877** |

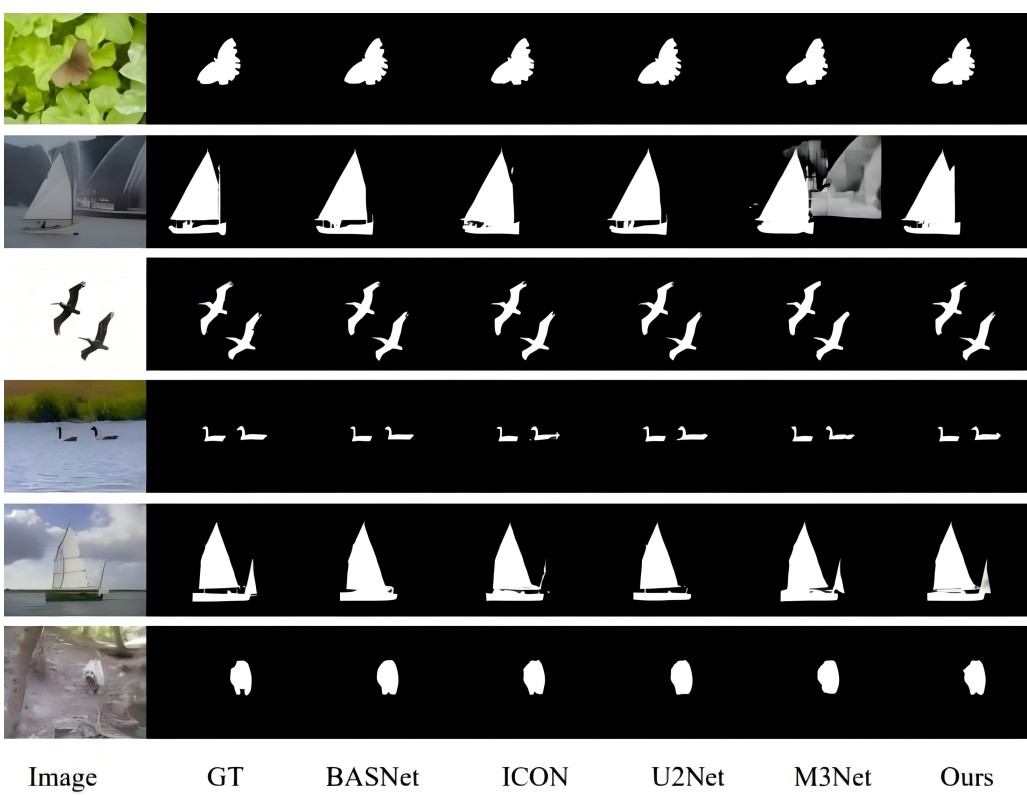

| Image | GT | BASNet | ICON | U2Net | M3Net | Ours |

**Figure 5  Qualitative comparison with state-of-the-art techniques, the proposed algorithm can detect more complete and significant targets with finer edges.**

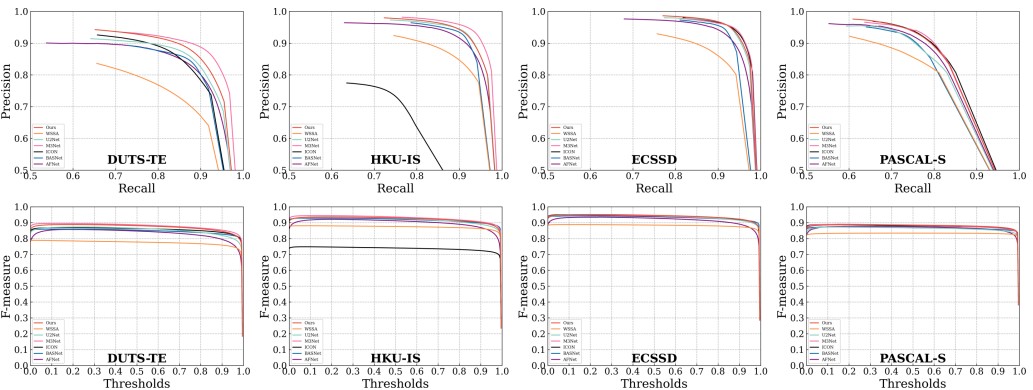

**Figure 6 Comparison with other algorithms on four popular SOD datasets.** The first two rows and the third four rows show the PR curve and the F-measure curve, respectively. It can be seen that the proposed method performs better.

Notably, in both the PR curves and F-measure curves, the solid red line representing our proposed method consistently outperforms all other methods across most thresholds. This superiority can be attributed to the integration of our proposed SGPFMs, which facilitate intra-convolutional layer self-optimization. This capability effectively mitigates issues related to predicting saliency graphical target errors and incomplete saliency prediction. Furthermore, our model leverages the BFF modules, which engender cross-optimization between saliency target features and saliency edge features. Consequently, our model is adept at producing sharp foreground target edges while suppressing background interference in edge features.

We show the feature maps generated by the entire network during training. All feature maps are obtained by dimensionality reduction of the corresponding features. The images go through Resnet50 input to SGPFM for rough features, F5 output for edge features and F5 output for saliency features. After being processed by the SGPFM module and the BFF module, the background interference is effectively suppressed, and the edge features and saliency features of the foreground targets (*e.g.*, the car and the bird) are presented as clear contours in the output of BFF1. For both edge features and saliency features, the predicted salient target segmentation results are more accurate and the prediction results of salient edges have more coherent contours due to the cross-optimization of these two kinds of information in multiple BFF modules. The improved results presented in Fig. 7 show that both segmentation performance and edge detection performance are gradually optimized when BSEFNet is equipped with multiple BFF modules.

## Ablation analysis

An exhaustive ablation study was conducted to ascertain the efficacy of the different modules proposed in this paper. The results are tabulated in Table 2. We systematically replaced the DASSP module, asymmetric convolution (AC) and self-attention mechanisms within the SGPFMs, and the reverse convolution and feature fusion module (Fused Future + Deconv, FD) at the network's output with simpler channel and reduction operations.

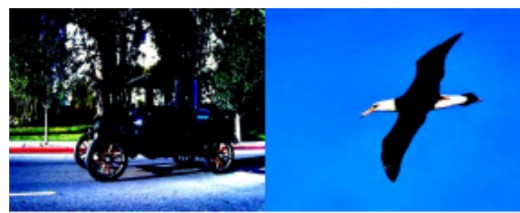

Image

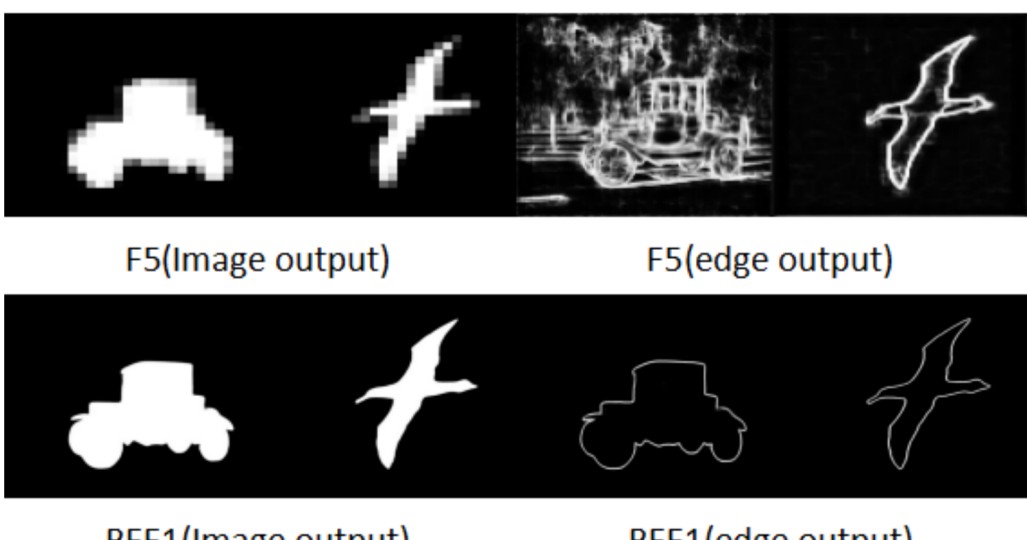

F5(Image output)      F5(edge output)

BFF1(Image output)      BFF1(edge output)

**Figure 7** **Comparison of saliency feature map and saliency edge feature map, where (image output) is the saliency feature of the output and (edge output) is the edge feature of the output.**

**Table 2** **Using ResNet as a baseline, different components are added incrementally to validate their effectiveness.**

| ResNet DASSP | AC | Self-Attention | FD | DUTS-TE MaxF ↑ | MeanF ↑ | MAE ↓ |
|---|---|---|---|---|---|---|
| ✓ | | | | 0.866 | 0.851 | 0.042 |
| ✓ | ✓ | | | 0.872 | 0.86 | 0.04 |
| ✓ | ✓ | ✓ | | 0.875 | 0.858 | 0.036 |
| ✓ | ✓ | ✓ | ✓ | 0.888 | 0.876 | 0.035 |

Notably, each module introduced notable performance enhancements over the benchmark ResNet model, thus affirming their effectiveness. Moreover, the performance exhibited a progressive improvement with the incremental addition of these components. The consistent performance gains underscore the synergistic interplay among the proposed modules and their collective effectiveness in maximizing saliency detection performance (as indicated in the last row). Notably, the overall gain over the baseline model on the DUTS-TE dataset escalated by 2.5%.

## CONCLUSION

In this paper, we propose a new saliency target detection model, BSEFNet, which effectively combines saliency semantic features and edge features. We design SGPFM to improve the feature extraction capability of the detected objects, effectively fusing saliency and edge features, reducing the cases of missing objects and improving the boundaries of the objects. We transfer the grouped features output from SGPFM to the BFF module for bi-directional optimization, which enhances the representation of intra-layer features. Compared with some state-of-the-art algorithms, our proposed BSEFNet achieves better performance on the four underlying datasets.

### Funding

This research was funded by the National Natural Science Foundation of China (Grant No. 62301473), the Jiangsu Higher Education Institutions' Brand Specialty Construction Project, the Jiangsu Province Graduate Research and Practice Innovation Program (SJCX23 1871, SJCX24 2152, SJCX24 2153), the Jiangsu Province Higher Education Teaching Reform Research Project Fund (Grant No. 2023JSJG399), the Jiangsu Province First-Class Discipline Construction Project (TAPP), the 2023 Jiangsu Province Industry-University-Research Cooperation Project: Research on the Control System of the Flat Laminate Composite Material Production Line (Grant No. BY20231069), the Jiangsu Province Industry-University-Research Cooperation Project (Grant No. BY20231068), the Jiangsu Province Higher Education "High-Quality Public Course Teaching Reform Research" Support Project (Grant No. 2022JDKT110), and the Jiangsu Province Higher Education "Electronic Information Specialty Construction, Course Construction, and Teaching Research" Projects (Grant Nos. 2024JSDZJG25, 2024JSDZJG59). The funders had no role in study design, data collection and analysis, decision to publish, or preparation of the manuscript.

### Grant Disclosures

The following grant information was disclosed by the authors:
The National Natural Science Foundation of China:  62301473.
The Jiangsu Higher Education Institutions' Brand Specialty Construction Project.
The Jiangsu Province Graduate Research and Practice Innovation Program: SJCX23 1871, SJCX24 2152, SJCX24 2153.
The Jiangsu Province Higher Education Teaching Reform Research Project Fund: 2023JSJG399.
The Jiangsu Province First-Class Discipline Construction Project (TAPP).
The 2023 Jiangsu Province Industry-University-Research Cooperation Project: Research on the Control System of the Flat Laminate Composite Material Production Line: BY20231069.
the Jiangsu Province Industry-University-Research Cooperation Project: BY20231068.
The Jiangsu Province Higher Education "High-Quality Public Course Teaching Reform Research": 2022JDKT110.

The Jiangsu Province Higher Education "Electronic Information Specialty Construction, Course Construction, and Teaching Research" Projects: 2024JSDZJG25, 2024JSDZJG59.

## Competing Interests

The authors declare there are no competing interests.

## Author Contributions

- Gan Gao conceived and designed the experiments, performed the experiments, analyzed the data, performed the computation work, prepared figures and/or tables, authored or reviewed drafts of the article, and approved the final draft.
- Yuanyuan Wang analyzed the data, prepared figures and/or tables, authored or reviewed drafts of the article, and approved the final draft.
- Feng Zhou analyzed the data, prepared figures and/or tables, authored or reviewed drafts of the article, and approved the final draft.
- Shuaiting Chen conceived and designed the experiments, performed the experiments, analyzed the data, performed the computation work, prepared figures and/or tables, authored or reviewed drafts of the article, and approved the final draft.
- Xiaole Ge performed the experiments, analyzed the data, prepared figures and/or tables, authored or reviewed drafts of the article, and approved the final draft.
- Rugang Wang analyzed the data, prepared figures and/or tables, and approved the final draft.

## Data Availability

The BSEFNet code is available at Zenodo: Gao, G. (2024). BSEFNet [Zenodo]. https://doi.org/10.5281/zenodo.12697436.

The DUTS dataset including the DUTS-TR training dataset and the DUTS-TE testing dataset is available at http://saliencydetection.net/duts.

The HKU-IS Saliency Detection Dataset is available at: https://aistudio.baidu.com/datasetdetail/69843.

The Extended Complex Scene Saliency Dataset (ECSSD) is available at: https://www.cse.cuhk.edu.hk/leojia/projects/hsaliency/dataset.html.

The PASCAL-S Visual Object Classes Challenge 2010 (VOC2010) is available at: http://host.robots.ox.ac.uk/pascal/VOC/voc2010/index.html.

BASNet is available at GitHub: https://github.com/xuebinqin/BASNet.

ICON is available at GitHub: https://github.com/mczhuge/ICON?tab=readme-ov-file.

U2Net is available at GitHub: https://github.com/xuebinqin/U-2-Net.

M3Netis available at GitHub: https://github.com/Ayews/M3Net.

The BASNet predicted saliency maps are available at Zenodo: Gao, G. (2024). The BASNet predicted saliency maps. Zenodo. https://doi.org/10.5281/zenodo.13979729.

The U2Net predicted saliency maps are available at Zenodo: Gao, G. (2024). U2Net predicted saliency maps. Zenodo. https://doi.org/10.5281/zenodo.13979797.

The M3Net predicted saliency maps are available at Zenodo: Gao, G. (2024). M3Net predicted saliency maps. Zenodo. https://doi.org/10.5281/zenodo.13979820.

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
