# Peer review of "BSEFNet: bidirectional self-attention edge fusion network salient object detection based on deep fusion of edge features"

_PeerJ Computer Science, doi:10.7717/peerj-cs.2494_

## Round 0.1 · original submission · Major Revisions

Please address the changes and resubmit.

Reviewer 1 ·

Basic reporting

The code in zendo does not include all of the files.
The line "from model.resnet_daspp import ResNet_DASPP" can not be run.

The author said that they compare many methods:"These benchmark techniques comprise
255 AFNet(Feng et al., 2019b), BASNet (Qin et al., 2019b), MINet (Pang et al., 2020b), U2Net(Qin et al.,
256 2020), ITSD(Zhou et al., 2020), DSRNet(Wang et al., 2020), WSSA(Zhang et al., 2020), ICON-R(Zhuge
257 et al., 2022), M3Net-R(Yuan et al., 2023), and GFINet(Zhu et al., 2023), as delineated in Table 1.
"

But they are not in the code.

Experimental design

In line 241, the author only divides the data set into training and testing parts. How did the author select the epoch to stop training? Usually, the datasets are split into training/validation/testing sets. In the training process, the training set is used to train the model, and the validation set is used to tune the parameter and stop the condition. The test set should be set aside until final testing. I do not find this procedure in the manuscript.

Validity of the findings

The result may be overestimated if the test set is used in the training process. I can not evaluate the result if this information is not clearly provided.

Reviewer 2 ·

Basic reporting

No comment

Experimental design

No comment

Validity of the findings

No comment

Additional comments

My evaluations of the paper are below:
- The literature on the subject has not been adequately researched. Studies conducted in 2022 and later were not examined.
- A lot of detail is given in the introduction, but motivation and the contribution of the work are not well defined. Therefore, the contribution of the study to science is uncertain.
- The experimental results obtained are insufficient.
- The conclusion section of the study is careless.
In summary, this paper does not comply with SCI journal criteria.

Reviewer 3 ·

Basic reporting

- In abstract, you should focus on the key characteristics/developments/method to demonstrate briefly
- Introduction is too long, you should shorten with meaningful sentences
- Related works is too short, you must provide an overview of background works for readers to understand

Experimental design

- There should have a flowchart/diagram to demonstrate the routine of experimental setup
- Some experiences of developers could be described here

Validity of the findings

- The novelty and impact are not clear. You must clarify them in the body text

Additional comments

+ References are so many in this manuscript. However, it still lacks of several key researches. It is better for you to cite by yourself
+ Some of references are too old. I suggest that you should focus on recent publications (not over 5 years from present) in order to ensure the up-to-date technique

Reviewer 4 ·

Basic reporting

1. The authors claimed that existing algorithms have issues with saliency edge inaccuracies. However, their method still uses feature extraction from ResNet50. Though they have appended other algorithms over these features, they need to mathematically justify this. How the authors claim this approach works on other variants of CNN too?
2. As DASPP plays an important role in the study, cover it in the abstract. Include statistical result in Abstract.

Experimental design

In Line 225, authors have written “Empirically, they are set to o = 0.8 and λ1 = λ2 = 0.6.”. Why? Show results on different values.

Validity of the findings

1. Justify figure 4, second row, 6th column image
2. What about comparing the segmentation results with the ground truth?

Reviewer 5 ·

Basic reporting

All comments have been added in detail to the last section.

Experimental design

All comments have been added in detail to the last section.

Validity of the findings

All comments have been added in detail to the last section.

Additional comments

Review Report for PeerJ Computer Science
(BSEFNet: salient object detection based on deep fusion of edge features)

1. Within the scope of the study, a deep learning based model called BSEFNet, which can perform salient object detection operations using multiple popular and open source image datasets, is proposed.

2. In the Introduction section, the purpose of salient object detection, traditional and current salient object detection problems, and the main contributions of this study to the literature are clearly and sufficiently mentioned.

3. In the Related works section, salient object detection studies in the literature are mentioned. This section definitely needs to be detailed. In particular, the studies carried out with the datasets used in the study can be given in more detail, and their fundamental differences from this study etc. can be mentioned.

4. When the structure of the proposed BSEFNet model is examined, it is observed that it consists of encoder and decoder parts, and also contains various bidirectional feature fusion modules and self attention group pixel fusion modules, dense spatial pyramid pooling model and ResNet. When the model details are examined, it is understood that it contains a certain level of originality, but there are parts that need to be explained and detailed. It is seen that there are many different deep learning models that can be used especially for the ResNet50 part of the proposed model, which is examined in the literature. For this reason, it should be explained in more detail why this was preferred and/or whether different experiments were performed. Similarly, please detail the reason for the preferred loss function and what are its differences from other loss functions that can be used in other salient object detection problems in the literature.

5. It is observed that the datasets used with the parameters given in the experimental setup section are at a sufficient level with the dataset diversity. In addition, the results obtained with the performance metric types are positive and very suitable for the study when compared to the literature.

As a result, although the study has the potential to make a significant contribution to the literature in the field of salient object detection, it is recommended to pay attention to the sections mentioned above.

---

## Round 0.2 · Minor Revisions

Kindly do your best to address the manuscript as per the reviewer 4 suggestion and resubmit it.

Reviewer 1 ·

Basic reporting

The issues have been improved.

Experimental design

The issues have been improved.

Validity of the findings

The issues have been improved.

Reviewer 2 ·

Basic reporting

No comment

Experimental design

No comment

Validity of the findings

No comment

Additional comments

By reviewing all the points written by the referees in the previous revision, the paper has been made more readable and understandable. I thank the authors for their efforts. The paper can be accepted as it is.

Reviewer 3 ·

Basic reporting

It can be accepted

Experimental design

It can be accepted

Validity of the findings

It can be accepted

Reviewer 4 ·

Basic reporting

Still, I am not satisfied with my first comment. "Planning to do" is not the better answer.

Experimental design

N/A

Validity of the findings

N/A

Additional comments

N/A

Reviewer 5 ·

Basic reporting

All comments have been added in detail to the last section.

Experimental design

All comments have been added in detail to the last section.

Validity of the findings

All comments have been added in detail to the last section.

Additional comments

Review Report for PeerJ Computer Science
(BSEFNet: Bidirectional Self-Attention Edge Fusion Network salient object detection based on deep fusion of edge features)

When the responses given and the related changes in the paper are examined in detail, it is observed that they are at an adequate level. From this perspective, I recommend that the paper be accepted.

---

## Round 0.3 · accepted · Accept

Manuscript is accepted for publication.

Reviewer 4 ·

Basic reporting

All my queries are answered.

Experimental design

N/A

Validity of the findings

N/A

Additional comments

N/A